

# A global view of the off-shell Higgs portal

Maximilian Ruhdorfer⋆, Ennio Salvioni and Andreas Weiler

Physik-Department, Technische Universität München, 85748 Garching, Germany

⋆ max.ruhdorfer@tum.de

## Abstract

We study for the first time the collider reach on the *derivative Higgs portal*, the leading effective interaction that couples a pseudo Nambu-Goldstone boson (pNGB) scalar Dark Matter to the Standard Model. We focus on Dark Matter pair production through an off-shell Higgs boson, which is analyzed in the vector boson fusion channel. A variety of future high-energy lepton colliders as well as hadron colliders are considered, including CLIC, a muon collider, the High-Luminosity and High-Energy versions of the LHC, and FCC-hh. Implications on the parameter space of pNGB Dark Matter are discussed. In addition, we give improved and extended results for the collider reach on the *marginal Higgs portal*, under the assumption that the new scalars escape the detector, as motivated by a variety of beyond the Standard Model scenarios.

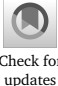
# 1 Introduction

The hypothesis that Dark Matter (DM) is composed of weakly interacting massive particles (WIMPs) whose abundance is determined by thermal freeze-out has been a leading paradigm for decades. The direct searches for DM scattering on nuclei, however, have kept reporting null results, lowering the cross section limits at an impressive pace and ruling out many WIMP models along the way. Currently, the strongest sensitivity has been achieved by the XENON1T experiment [1].

A compelling exception to this tension is obtained if the WIMP arises as a pseudo Nambu-Goldstone boson (pNGB). This possibility is especially motivated by theories where the Higgs and the DM arise together as composite pNGBs, thus addressing simultaneously the naturalness and DM puzzles [2]. Models of this type have received increasing attention lately [3–17]. The key point is that the scalar DM, assumed to be a singlet under the Standard Model (SM) gauge interactions, couples to the SM through higher-dimensional, derivative interactions with the Higgs field, that arise from the kinetic term of the nonlinear sigma model. Up to dimension six the relevant Lagrangian contains the *derivative Higgs portal*,

$$\mathcal{L}_{\text{derivative}} = \mathcal{L}_{\text{SM}} + \frac{1}{2}(\partial_\mu \phi)^2 - \frac{1}{2}m_\phi^2 \phi^2 + \frac{c_d}{2f^2}\partial_\mu \phi^2 \partial^\mu |H|^2, \tag{1}$$

where $\phi$ is the scalar DM candidate and $H$ the Higgs doublet. The scale $f$ is the common decay constant of the pNGBs, while $c_d$ is an $O(1)$ coefficient. For definiteness we have taken $\phi$ to be a real scalar stabilized by a $Z_2$ symmetry. The important observation is that the operator proportional to $c_d$ mediates $s$-wave annihilation to SM particles, but is very suppressed in the scattering of DM on nuclei, which is characterized by low momentum transfer $|q| \lesssim 100$ MeV. Hence, thermal freeze-out can yield the observed DM density for $m_\phi \sim O(100)$ GeV and $f \sim O(1)$ TeV, while the direct detection signal is beyond the foreseeable experimental reach. In complete models additional terms can be present beyond Eq. (1), which either respect or break explicitly the DM shift symmetry (some explicit breaking is of course necessary, to generate a nonvanishing $m_\phi$), but the derivative Higgs portal is the irreducible ingredient underlying this type of DM candidate.

A realization of pNGB DM can also be obtained [18] by adding a complex scalar field to the SM with potential invariant under a $U(1)$, which is softly broken only by a mass term [18,19] (notice that in this setup the Higgs field is not a pNGB). The angular mode of the scalar is stabilized by a remnant $Z_2$ and is identified with the pNGB DM. As we briefly discuss in Sec. 2, if the radial mode is sufficiently heavy it can be integrated out, yielding Eq. (1) as the low-energy theory. See e.g. Refs. [20–34] for subsequent studies of models built following this approach.

Experimental probes of the derivative Higgs portal include indirect DM searches in cosmic rays, and collider experiments. In this work we explore the latter for the first time, by considering the production of two invisible $\phi$ particles through an $s$-channel Higgs boson.[1] While for $m_\phi < m_h/2$ the sensitivity on $f/c_d^{1/2}$ is immediately obtained from the available studies of invisible Higgs decays, for $m_\phi > m_h/2$ the Higgs is off-shell and the momentum dependence of the coupling in Eq. (1) has an important impact on the signal kinematic distributions, mandating a dedicated study that is the main subject of this paper. We focus on Higgs production via vector boson fusion (VBF), see Fig. 1, because it is expected to provide the best sensitivity both at high-energy lepton colliders and at hadron colliders. We will expand on this momentarily.

---

[1]Reference [35] considered this process in the monojet channel at the LHC, but no results were provided for the theory in Eq. (1). An extended model with momentum-dependent couplings was considered instead, finding increased sensitivity in comparison to the momentum-independent case. Our analysis of the vector boson fusion channel confirms this behavior, see in particular Fig. 8.

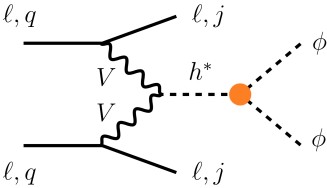

Figure 1: Feynman diagram for the signal studied in this work. The orange dot represents either of the portals in Eqs. (1) and (2).

Throughout our analysis we also consider the *marginal* (or *renormalizable*) *Higgs portal* [36–38],

$$\mathcal{L}_{\text{marginal}} = \mathcal{L}_{\text{SM}} + \frac{1}{2}(\partial_\mu \phi)^2 - \frac{1}{2}M_\phi^2 \phi^2 - \frac{\lambda}{2}\phi^2 |H|^2 \,, \qquad (2)$$

which provides an important term of comparison.[2] We assume that $\phi$ escapes the detector and thus manifests as missing momentum. While the hypothesis that $\phi$ is a thermal relic that interacts with the SM through Eq. (2) is mostly ruled out by direct detection (exceptions being the Higgs resonance region, or DM heavier than a few TeV) [39], non-standard but motivated cosmological histories can open up large regions of parameter space [40]. Furthermore, new scalars with sizable couplings to the Higgs are broadly motivated by open problems of the SM other than DM, such as baryogenesis and naturalness. These scalars may be stable on collider timescales or decay invisibly, giving rise to the signature studied here. This scenario was discussed in the context of electroweak baryogenesis in Ref. [41]. Furthermore, natural models where the top partners are scalars require that they couple to the Higgs with strength fixed by $y_t^2$. If the scalar top partners are neutral under all SM gauge symmetries, as in the recently-proposed Neutral Naturalness theories of Refs. [42,43], then probing the interactions $\mathcal{L} \ni -y_t^2|H|^2(|\tilde{u}_1^c|^2 + |\tilde{u}_2^c|^2)$ through an off-shell Higgs could be key to discovering this type of solution to the little hierarchy problem.

Previous studies of the off-shell marginal Higgs portal to invisible scalars are available in the literature. The sensitivity on $\lambda$ at lepton colliders was carefully analyzed in Ref. [44] for $\sqrt{s} \leq 1$ TeV, where $Zh$ associated production mostly dominates (see also Refs. [45, 46] for related studies), and in Ref. [47] for $\sqrt{s} = 1, 5$ TeV, considering $ZZ$ fusion production.[3] The sensitivity at the LHC and FCC-hh was thoroughly examined in Ref. [48], including the VBF, monojet and $t\bar{t}h$ production modes.

Here we focus on VBF production, which at lepton colliders provides the leading sensitivity for $\sqrt{s} \gtrsim 1$ TeV, and at hadron colliders was found to be superior to monojet and $t\bar{t}h$ in the previous study of Ref. [48]. We provide the first results for the derivative Higgs portal, in the form of sensitivity projections for a wide set of colliders that include both high-energy lepton machines and hadron machines. The collider parameters we have assumed are summarized in Table 1. We also extend or revisit previous findings for the marginal Higgs portal:

Table 1: Collider parameters used in our off-shell Higgs projections.

|  | HL-LHC | CLIC 1.5 | HE-LHC | CLIC 3 | FCC 100 | $\mu$C 6 | $\mu$C 14 |
|---|---|---|---|---|---|---|---|
| Center of mass energy [TeV] | 14 | 1.5 | 27 | 3 | 100 | 6 | 14 |
| Integrated luminosity [ab$^{-1}$] | 3 | 1.5 | 15 | 3 | 30 | 6 | 14 |

---

[2]In this case the physical mass of the scalar is $m_\phi^2 = M_\phi^2 + \lambda v^2/2$, where $v \simeq 246$ GeV.

[3]At lepton colliders $WW$ fusion leads to an undetectable final state if $\phi$ is invisible, hence one must rely on $ZZ$ fusion.

on the lepton collider front, we consider higher-energy proposals such as CLIC and a muon collider; on the hadron collider front, we perform an updated analysis that includes improved background predictions, as well as the impact of trigger thresholds on the missing transverse energy requirements. We also include projections for the HE-LHC. Our VBF analysis is pre-

Table 2: 95% CL exclusion limits obtained from our off-shell Higgs analysis. See Sec. 3 for details.

|  | HL-LHC | CLIC 1.5 | HE-LHC | CLIC 3 | FCC 100 | $\mu$C 6 | $\mu$C 14 |
|---|---|---|---|---|---|---|---|
| derivative, $m_\phi = 100$ GeV: $f/c_d^{1/2}$ [GeV] | 280 | 280 | 450 | 540 | 880 | 980 | 2000 |
| marginal, $\lambda = \sqrt{4N_c}\, y_t^2$ : $m_\phi$ [GeV] | 130 | 170 | 190 | 310 | 330 | 540 | 990 |

sented in Sec. 3, but we anticipate its main results in Fig. 2, which provides an overview of the sensitivity on both types of portals. The reach along benchmark slices of the parameter space is summarized in Table 2. In the presence of $N$ real scalars, degenerate and with identical couplings, their effect on the signal studied in this paper is equivalent to that of a single field with $\{c_d, \lambda\} \to \sqrt{N}\,\{c_d, \lambda\}$. For example, for complex pNGB DM $\chi$ with derivative portal $\partial_\mu |\chi|^2 \partial^\mu |H|^2/f^2 \in \mathcal{L}$ we should take an effective interaction strength $c_d = \sqrt{2}$, whereas for degenerate scalar top partners (two complex scalars in the $SU(N_c)$ fundamental, with $y_t^2$ coupling to the Higgs) we have $\lambda = \sqrt{4N_c}\, y_t^2$. Note also that if the integrated luminosity turns out to be $L'$ rather than the assumed $L$, the reach on $f/c_d^{1/2}$ scales as $(L'/L)^{1/8}$ and the reach on $\lambda$ scales as $(L/L')^{1/4}$ (this neglects systematic uncertainties, on which we will comment at the end of Sec. 3).

Before concluding this Introduction, we briefly summarize the reach for $m_\phi < m_h/2$, in which case *on-shell* invisible Higgs decays are the main search avenue. These have been analyzed in great detail in the literature, and the resulting bounds on the Higgs branching ratio to new invisible particles[4] are reported in the upper row of Table 3. At the LHC the strongest sensitivity is obtained in VBF production [49], see e.g. Refs. [50,51] for early analyses. A future Higgs factory will increase the reach significantly, by exploiting $Zh$ production, and an even stronger bound can be achieved at FCC-hh. The analysis of on-shell decays is insensitive to the specific portal considered, and the limits on BR($h \to$ inv) are straightforwardly translated into constraints on $f/c_d^{1/2}$ and $\lambda$ by using the expression of the width,

$$\Gamma(h \to \phi\phi) = \frac{v^2}{32\pi m_h}\Big(\lambda - c_d \frac{m_h^2}{f^2}\Big)^2 \Big(1 - \frac{4m_\phi^2}{m_h^2}\Big)^{1/2}. \tag{3}$$

The results are given in the middle and bottom rows of Table 3.

The remainder of this paper is organized as follows. In Sec. 2 we shortly discuss the pNGB DM framework, using a general effective field theory (EFT) for the SM plus the scalar DM. Section 3 presents our analysis of the off-shell derivative and marginal Higgs portals, exploiting the VBF channel at future colliders. We discuss our results and give an outlook in Sec. 4. Appendix A provides details on the analysis, including a summary of all the selection cuts we imposed and an explicit comparison with the results of Ref. [48]. Finally, Appendix B gives general cross sections for pNGB DM scattering on nuclei and annihilation.

---

[4]Notice that the SM predicts an invisible Higgs branching fraction of BR($h \to ZZ^* \to 4\nu)_{\rm SM} \approx 1.1 \cdot 10^{-3}$.

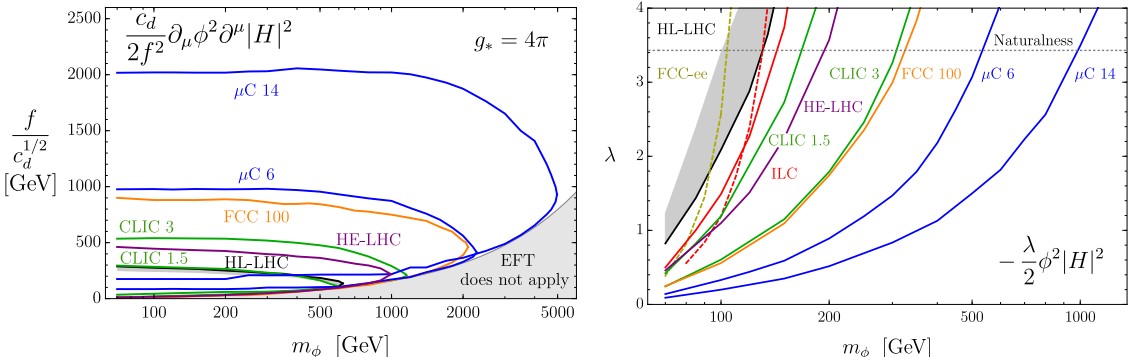

Figure 2: Projected 95% CL constraints on the Higgs portal couplings at future colliders. *Left:* derivative Higgs portal. To remain within the regime of validity of the EFT we only retain events with $M_{\phi\phi} < g_* f / c_d^{1/2}$, where $g_*$ is set to $4\pi$. This requirement cannot be satisfied in the gray region $2m_\phi > g_* f / c_d^{1/2}$. The excluded region is the one enclosed by each line. All limits are derived from VBF. *Right:* marginal Higgs portal. The excluded region is the one above each line. Solid lines correspond to VBF processes: in addition to the benchmarks in Table 1, we show ILC 1 TeV with 1 ab$^{-1}$ (solid red). Dashed lines correspond to $Zh$ production from Ref. [44]: ILC 500 GeV with 1 ab$^{-1}$ (dashed red) and FCC-ee at 350 GeV and 2.5 ab$^{-1}$ (dashed yellow). The dotted line labeled "Naturalness" corresponds to $\lambda = \sqrt{4N_c}\, y_t^2$, namely two degenerate scalar top partners. In both panels, for the HL-LHC we also show a gray band whose weaker limit boundary corresponds to adding a 1% systematic uncertainty on the background.

Table 3: 95% CL exclusion limits on the coefficients of the derivative and marginal Higgs portal operators, obtained from *on-shell* invisible Higgs decays, assuming $m_\phi \ll m_h/2$. See text for details.

|  | LHC current[52] | HL-LHC | ILC 250[44] | FCC-ee 240[44] | FCC-hh[55] |
|---|---|---|---|---|---|
|  | (mostly VBF) | VBF[53] [$Zh$][54] | ($Zh$) | ($Zh$) | (inclusive) |
| BR($h \to$ inv) | 0.19 | 0.035 [0.08] | $1.3 \cdot 10^{-3}$ | $8 \cdot 10^{-4}$ | $2.5 \cdot 10^{-4}$ |
| $f / c_d^{1/2}$ [TeV] | 1.0 | 1.7 [1.3] | 3.8 | 4.3 | 5.8 |
| $\lambda$ [$10^{-2}$] | 1.4 | 0.55 [0.86] | 0.10 | 0.082 | 0.046 |

## 2 Pseudo Nambu-Goldstone boson Dark Matter

We focus on theories where both the Higgs doublet and a scalar DM candidate, singlet under the SM gauge symmetries, arise as light pNGBs from a strongly coupled sector. In this section we take the DM to be a complex scalar $\chi$; for real DM $\phi$, one simply replaces $\chi \to \phi/\sqrt{2}$. We consider the effective Lagrangian

$$\mathcal{L} = \mathcal{L}_{\text{SM}} + |\partial_\mu \chi|^2 - m_\chi^2 |\chi|^2 + \mathcal{L}_{\text{int}}, \tag{4}$$

where the DM-SM interactions are described in general, up to dimension six, by

$$\mathcal{L}_{\text{int}} = \frac{c_d}{f^2} \partial_\mu |\chi|^2 \partial^\mu |H|^2 + \frac{i}{f^2} (\chi^* \overset{\leftrightarrow}{\partial_\mu} \chi) \sum_{\Psi = q_L, u_R, d_R, \ell_L, e_R} b_\Psi \overline{\Psi} \gamma^\mu \Psi + \frac{ig'}{m_*^2} c_B (\chi^* \overset{\leftrightarrow}{\partial_\mu} \chi) \partial_\nu B^{\mu\nu}$$

$$- \lambda |\chi|^2 |H|^2 \tag{5}$$

$$+ \frac{|\chi|^2}{f^2} \left( c_u^\chi y_u \overline{q}_L \tilde{H} u_R + c_d^\chi y_d \overline{q}_L H d_R + c_e^\chi y_e \overline{\ell}_L H e_R + \text{h.c.} \right) + \sum_{V = G, W, B} \frac{d_V y^2}{16\pi^2} \frac{g_V^2}{m_*^2} |\chi|^2 V^{\mu\nu} V_{\mu\nu}.$$

Here $\chi^* \overset{\leftrightarrow}{\partial_\mu} \chi \equiv \chi^* \partial_\mu \chi - \chi \partial_\mu \chi^*$ and $m_* = g_* f$ with $g_*$ a coupling, while $g_{G,W,B} = g_s, g, g'$. A sum over the fermion generations is understood. The operators in the first line preserve the DM shift symmetries,[5] whereas those in the second and third lines explicitly break them (in the last term, $y$ generically indicates the relevant shift symmetry breaking coupling). We assume $CP$ and custodial invariance: the former implies that all the coefficients in Eq. (5) are real, whereas the latter forbids the operator $(\chi^* \overset{\leftrightarrow}{\partial_\mu} \chi)(H^\dagger \overset{\leftrightarrow}{D_\mu} H)$. Note that we have neglected $|\chi|^2 |H|^4$, since for our purposes it only gives a subleading contribution to $\lambda$, with relative suppression $\sim v^2/f^2$. The Lagrangian in Eq. (5) is equivalent to the one presented in Refs. [57,58], with a different choice for the operator basis. Our normalization is such that the $b, c$ and $d$ coefficients have maximal size of $O(1)$, according to the SILH power counting [56].[6] They can, however, be smaller, for example due to additional symmetries, or because they are suppressed by the degree of compositeness of the SM fermions. Examples of both situations will be discussed momentarily. In Eq. (5) we have neglected additional operators that are expected on general grounds, but do not involve the DM. Among these the operator $c_H (\partial_\mu |H|^2)^2/(2f^2)$ is especially important, because it causes a rescaling of all the Higgs couplings by $1 - O(c_H v^2/f^2)$.

For $m_\chi \gtrsim 10$ GeV, the null results of direct searches for DM scattering on nuclei set constraints on the EFT parameters. Only a few directions are strongly constrained. Using the cross section reported in Eq. (21) from Appendix B and taking as an example $m_\chi = 100$ GeV, the latest XENON1T bound $\sigma_{\chi N} \lesssim 10^{-46}$ cm$^2$ (at 90% CL) [1] translates to

$$\frac{f}{b_1^{1/2}} \gtrsim 56 \text{ TeV}, \qquad \frac{m_*}{c_B^{1/2}} \gtrsim 6.5 \text{ TeV}, \qquad \lambda \lesssim 1.1 \cdot 10^{-2}, \tag{6}$$

where in the first case we set $b_{q_L}^1 = b_{u_R}^1 = b_{d_R}^1 = b_1$ for the first generation, whereas in the last two cases we took one operator at a time. Naively, the first constraint in Eq. (6) appears to be the strongest. However, we should recall that the expected size of the $b$ coefficients is $b_\Psi \sim \epsilon_\Psi^2$, where $\epsilon_\Psi$ is the compositeness fraction of the corresponding fermion field, which is typically very small except for the quarks of the third generation: assuming comparable compositeness for the left and right fermion chiralities, we have $b_{q_L} \sim b_{\psi_R} \sim \sqrt{2} m_\psi/(g_* v)$, which plugging in the numerical values of $m_{u,d}$ gives $f \gtrsim O(100)$ GeV for any interesting $g_*$. Hence, this bound is easily irrelevant in realistic models. The constraint on $m_*/c_B^{1/2}$ in Eq. (6) (which is due to photon exchange between the DM and the quarks) is more interesting, as it pushes $g_*$ toward the fully strongly coupled regime for $f \sim$ TeV. However, we note that all operators of Eq. (5) that contain the DM current are absent if hidden-charge conjugation, which transforms $\chi \to -\chi^*$ but leaves the SM invariant, is preserved. This is the case for all the models presented in Ref. [15]. Furthermore, these operators vanish trivially for real

---

[5]Naively, the derivative Higgs portal operator may not seem shift-symmetric. This is just a consequence of our choice of basis for the Goldstone fields (see e.g. Ref. [15]), which is the same adopted in the Strongly Interacting Light Higgs (SILH) Lagrangian [56]. In this basis the shift invariance is not immediate.

[6]If the SM transverse gauge bosons are also composite [59] the enhancements $c_B \sim g_*/g'$ and $d_V \sim 16\pi^2/y^2$ are possible.

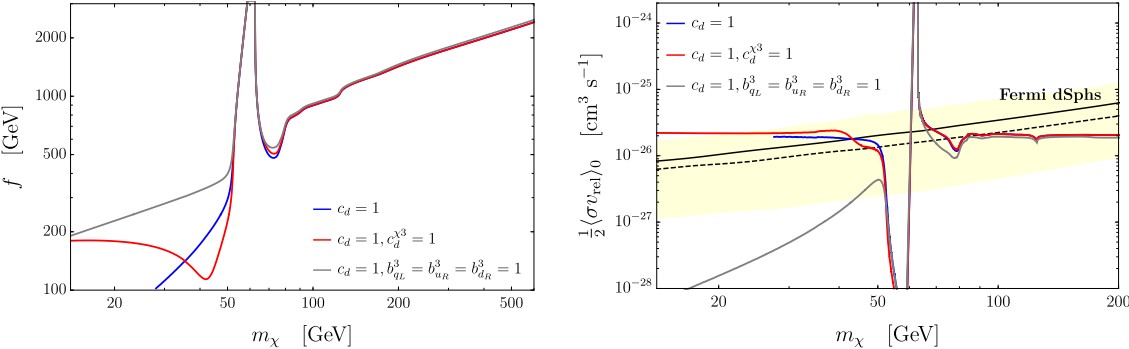

Figure 3: *Left:* contours obtained requiring that the thermal relic density of $\chi$ matches the observed one, assuming the given values of the EFT coefficients and setting the others to zero. *Right:* the present-day annihilation cross section to SM particles, calculated along the relic density contours shown in the left panel. The black line (yellow band) is the 95% CL observed upper limit (95% CL uncertainty band on the expected limit) from the dSphs analysis in Ref. [60]. The dashed black line corresponds to the observed limit from the analysis of a smaller dSphs sample [61]. The quoted experimental bounds assume annihilation to $b\bar{b}$.

scalar DM. More important is the constraint on $\lambda$ in Eq. (6), which has crucial implications for model building, even though this coupling only arises at loop level for pNGB DM: models where $\lambda$ is generated by top loops [2, 4, 12] have either been excluded or are presently being tested at XENON1T. However, recently Ref. [15] proposed several scenarios where $\lambda$ is very suppressed, either due to the smallness of the light SM fermion masses or because it arises at higher loop order.[7] In these models the DM-nucleon scattering cross section is well below the current bound, and may even be under the neutrino floor.

While our original motivation for the Lagrangian in Eqs. (4, 5) are theories where both the Higgs and the DM arise as composite pNGBs, this EFT can also apply to models such as that of Ref. [18], where a complex scalar $S$ is added to the SM as [18, 19, 23]

$$\mathcal{L} = \mathcal{L}_{\text{SM}} + |\partial_\mu S|^2 + \frac{\mu_S^2}{2}|S|^2 - \frac{\lambda_S}{2}|S|^4 - \lambda_{HS}|S|^2|H|^2 + \frac{\mu_S'^2}{4}(S^2 + \text{h.c.}). \tag{7}$$

The $U(1)$ symmetry acting on $S$ is broken spontaneously by the VEV $v_s$, where $S = (v_s + \sigma)e^{i\phi/v_s}/\sqrt{2}$, and explicitly by the term proportional to $\mu_S'^2$, yielding $\phi$ as a real pNGB DM candidate with mass $\mu_S'$, stabilized by the remnant $S \to S^*$ invariance. If the radial mode $\sigma$ is sufficiently heavy, it can be integrated out, obtaining an effective Lagrangian that after a field redefinition matches Eq. (5) with $c_d/f^2 = \lambda_{HS}/m_\sigma^2$.[8] In this setup $\lambda$ arises at one loop and results in a very suppressed cross section for DM-nucleon scattering [24, 25]. Gravitational wave signals from phase transitions in the early Universe have also been explored [28, 32]. Extended models of this class were discussed in Refs. [27, 29, 31].

In summary, there exists a class of WIMP models where the direct detection constraints are structurally satisfied due to the pNGB nature of the DM, and in particular $\lambda$ has negligible impact on the phenomenology. Depending on the specific realization, the DM-nucleus scattering signal may be within reach of future experiments, mediated for example by the contact

---

[7]Similarly, in these scenarios the operators proportional to $d_V$ are suppressed, either because $y$ is a light SM fermion Yukawa or because they arise at higher loops.

[8]Conversely, if the radial mode is relatively light it could be produced on-shell in VBF and decay to a DM pair, as studied in Ref. [26] at the LHC. Here we focus instead on DM production through the off-shell 125 GeV Higgs boson.

interactions between the DM and the down quarks parametrized by $c_d^\chi$, if the DM mass arises from bottom quark loops [15]. However, the signal may also be hidden below the neutrino floor, making its detection extremely challenging. The key element all these scenarios have in common is the derivative Higgs portal operator $c_d$, which is negligible in DM-nucleus scattering, but unsuppressed in annihilation. The observed relic density is reproduced via the freeze-out mechanism for $f \sim$ TeV, as shown in the left panel of Fig. 3. For completeness we show results down to $f = 100$ GeV, although it should be kept in mind that for weak-scale $f$, finding ultraviolet models that are acceptably described at low energies by Eq. (5) may be challenging.

Since $c_d$ mediates $s$-wave annihilation, indirect searches are crucial probes of pNGB DM. In the right panel of Fig. 3 we compare the present-day annihilation cross section to the bounds obtained by the Fermi-LAT [60,61] from gamma-ray observations of dwarf spheroidal galaxies (dSphs). For DM mass of $O(100)$ GeV, these bounds (which were computed assuming DM annihilation to $b\bar{b}$) are very close to the canonical thermal cross section $\langle \sigma v_{\rm rel}\rangle_{\rm can} \approx 2 \cdot 10^{-26}\,{\rm cm}^3\,{\rm s}^{-1}$. We see that the region $m_\chi \lesssim 70$ GeV is already in mild tension with data, and future improvements of the sensitivity will provide a stringent test of pNGB DM. Furthermore, searches for DM annihilation to monochromatic photons can be relevant for $m_h/2 \lesssim m_\chi \lesssim m_W$, where the branching ratio of the off-shell Higgs to $\gamma\gamma$ is $\gtrsim 10^{-3}$. We find that in this region the expected cross section is close to the Fermi-LAT sensitivity [62], if a favorable DM profile (contracted Navarro-Frenk-White) and the accordingly-optimized region of the sky are considered.

The recent measurement of the antiproton spectrum by AMS-02 [63] gives additional constraints, as well as intriguing hints of an excess [64,65]. Systematic uncertainties are larger in the antiproton channel, although they have been argued to be under control [66]. Recently, an explanation of the excess as originating from annihilations of pNGB DM has also been proposed [30]. As the situation has not settled yet, we do not show antiproton results here.

Clearly, should a robust excess emerge in indirect detection, its verification by other probes would be paramount to establish its origin. For pNGB DM, collider searches constitute the most important test, given the structural suppression of the direct detection cross section. The key ingredient of the scenario is the derivative Higgs portal, which can be probed in DM pair production as studied in this paper. Additional signatures may arise, depending on the values taken by the $b$ coefficients in Eq. (5). A theoretically motivated situation is one where the $b$'s are large only for the third generation quarks.[9] Then the process $gg \to t\bar{t} + \not{E}_T$ becomes relevant; at one loop, a contribution to monojet is also generated. The interplay of these two processes at the LHC was studied in Ref. [67] for fermionic DM. Here we limit ourselves to note that for scalar DM the $t\bar{t} \to \chi\chi^*$ amplitude at high energies is not suppressed by an $m_t$ insertion, plausibly leading to enhanced sensitivity.

## 3 The off-shell Higgs portal in vector boson fusion

We consider $\phi$ pair production via off-shell Higgs in VBF, which proceeds through the diagram in Fig. 1. The cross section is written as ($V = W$ or $Z$)

$$\sigma(f_1 f_2 \to \phi\phi f_1' f_2')[s] = \int_{4m_\phi^2/s}^{1} d\tau\, \hat{\sigma}_{VV\to\phi\phi}(\tau s)\, \mathcal{C}_{V_L V_L}^{f_1 f_2}(\tau), \qquad \tau \equiv \hat{s}/s, \tag{8}$$

---

[9]If the $b$'s are sizable for the first generation quarks, then important constraints arise from monojet searches [58]. However, as already mentioned, in this case direct detection rules out the entire $m_\chi \gtrsim 10$ GeV region.

where $\hat{s} = M_{\phi\phi}^2$. The parton luminosity is

$$\mathcal{C}_{V_L V_L}^{f_1 f_2}(\tau) = \int_\tau^1 \frac{dx}{x} f_{V_L/f_1}(x) f_{V_L/f_2}(\tau/x). \qquad (9)$$

The parton distribution functions (PDFs) for longitudinal polarizations, which are the only ones coupled to the Higgs, read in the limit $\hat{s} \gg m_V^2$ [68]

$$f_{V_L/f}(x) = \frac{(C_v^f)^2 + (C_a^f)^2}{4\pi^2} \frac{1-x}{x}, \qquad (10)$$

where for the $W$, $C_v^f = -C_a^f = g/(2\sqrt{2})$, and for the $Z$, $C_v^f = g_Z(T_L^{3f}/2 - s_w^2 Q^f)$ and $C_a^f = -g_Z T_L^{3f}/2$. Therefore

$$\mathcal{C}_{W_L W_L}(\tau) = \frac{g^4}{256\pi^4\tau}\left[2(\tau-1)-(\tau+1)\log\tau\right], \qquad \frac{\mathcal{C}_{Z_L Z_L}^{f_1 f_2}}{\mathcal{C}_{W_L W_L}} = \frac{R_{f_1}R_{f_2}}{4c_w^4} \equiv \mathcal{R}_w^{f_1 f_2}, \qquad (11)$$

where $R_f = 4(T_L^{3f})^2 - 8s_w^2 T_L^{3f} Q^f + 8s_w^4(Q^f)^2$. For the derivative and marginal Higgs portals, the partonic cross sections are

$$\hat{\sigma}_{VV\to\phi\phi}^{\rm deriv}(\hat{s}) = \frac{1}{32\pi} \frac{c_d^2 \hat{s}}{f^4}\left(1-\frac{m_h^2}{\hat{s}}\right)^{-2}\left(1-\frac{4m_\phi^2}{\hat{s}}\right)^{1/2},$$

$$\hat{\sigma}_{VV\to\phi\phi}^{\rm marg}(\hat{s}) = \frac{1}{32\pi} \frac{\lambda^2}{\hat{s}}\left(1-\frac{m_h^2}{\hat{s}}\right)^{-2}\left(1-\frac{4m_\phi^2}{\hat{s}}\right)^{1/2}, \qquad (12)$$

where we took the high-energy limit $\hat{s} \gg m_V^2$. Notice the relative enhancement of the derivative portal at high energy, $\hat{\sigma}^{\rm deriv}/\hat{\sigma}^{\rm marg} \propto \hat{s}^2$. Henceforth we describe our analysis, considering in turn high-energy lepton colliders and hadron colliders.

## 3.1 High-energy lepton colliders

At lepton machines the collision energy is equal to the collider center of mass energy, apart from the effect of initial state radiation, which we neglect in this paper. Assuming $\hat{s} \gg m_h^2$, we can perform the integral in Eq. (8) and obtain a simple analytical expression for the cross section. For the derivative coupling we find

$$\sigma(e^-e^+ \to \phi\phi e^-e^+) = \frac{\mathcal{R}_w^{e\bar{e}} g^4 c_d^2 s}{49152\pi^5 f^4}\left[\frac{3}{2} - \frac{2m_\phi^2}{s}\left(3\log^2\frac{s}{m_\phi^2} - 6\log\frac{s}{m_\phi^2} + 12 - \pi^2\right) + O(m_\phi^4/s^2)\right], \qquad (13)$$

where $\mathcal{R}_w^{e\bar{e}} \approx 0.11$ and we have expanded for $m_\phi^2/s \ll 1$. For the marginal portal one finds instead [69][10]

$$\sigma(e^-e^+ \to \phi\phi e^-e^+) = \frac{\mathcal{R}_w^{e\bar{e}} g^4 \lambda^2}{49152\pi^5 m_\phi^2}\left[\log\frac{s}{m_\phi^2} - \frac{14}{3} + \frac{m_\phi^2}{s}\left(3\log^2\frac{s}{m_\phi^2} + 18 - \pi^2\right) + O(m_\phi^4/s^2)\right]. \qquad (14)$$

Note the very different scalings with the collider energy and DM mass: for $s \gg m_\phi^2$ the derivative portal gives $\sigma \propto c_d^2 s/f^4$, which grows very quickly with $\sqrt{s}$ and is almost independent of

---

[10]Notice some typos in Ref. [69]: the second expression in Eq. (15) should have coefficient 256 instead of 4096, and the first expression should have 1/16 instead of 1/64. In addition, in the last expression of Eq. (14), $\hat{s}x/s \to \hat{s}/(sx)$. However, their numerical results are correct. We thank A. Tesi for correspondence about this point.

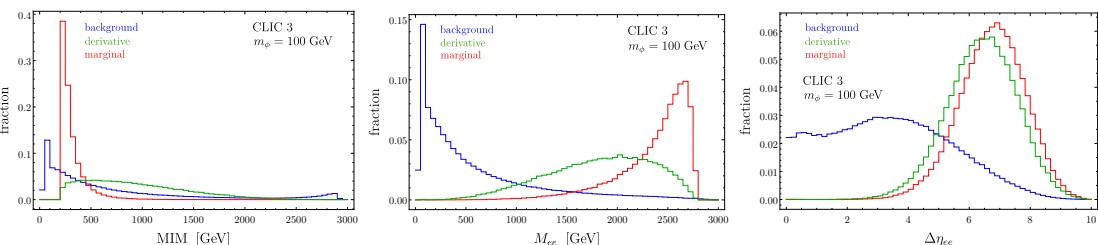

Figure 4: Normalized distributions for the signals and background at CLIC with $\sqrt{s} = 3$ TeV. The $\phi$ mass is set to 100 GeV.

$m_\phi$, whereas the marginal coupling leads to $\sigma \propto \lambda^2 \log(s/m_\phi^2)/m_\phi^2$, weakly dependent on the collider energy and rapidly decreasing as the DM mass is increased. Since the partonic cross section receives an important contribution from the threshold region $\hat{s} \approx 4m_\phi^2$, Eqs. (13) and (14) are accurate only if $m_\phi \gg m_h/2$. In our analysis we always use the exact cross sections as calculated by MadGraph5 [70] using a FeynRules 2.0 [71] implementation of Eqs. (1) and (2).

At lepton colliders we use MadGraph5 to perform a parton-level analysis, which is a reasonable approximation given the extremely clean final state consisting of two leptons and missing momentum. The dominant background is $e^- e^+ \to \nu \bar{\nu} e^- e^+$. At generation we require $p_T^e > 10$ GeV, $|\eta_e| < 5$ and $\Delta R_{ee} > 0.4$. The most useful kinematic variables to separate signal and background are: the missing transverse energy and missing invariant mass, defined as

$$\slashed{E}_T = (\slashed{p}_x^2 + \slashed{p}_y^2)^{1/2}, \qquad \text{MIM} = (\slashed{p}_\mu \slashed{p}^\mu)^{1/2}, \tag{15}$$

where $\slashed{p} = (\sqrt{s}, \vec{0}) - p_{e^-} - p_{e^+}$; as well as the invariant mass $M_{ee}$ and pseudorapidity separation $\Delta\eta_{ee} = |\eta_{e^-} - \eta_{e^+}|$ of the electron-positron pair. Distributions for the last three variables are shown in Fig. 4 for CLIC 3, chosen as representative example, and Table 4 describes the flow for the optimized cuts we adopt. For the derivative portal $M_{ee}$ provides less discriminating power compared to the marginal portal, but this is compensated by a tighter requirement on the MIM. After all cuts the background is dominated by $\nu_e \bar{\nu}_e e^- e^+$, including a contribution of $O(10)\%$ from on-shell $WW$ production.

Table 4: Cross sections in fb. For 3 ab$^{-1}$ we have $\mathcal{S} = 2.2\,[4.2]$ for the derivative [marginal] portal. No EFT consistency condition is applied.

| CLIC 3 | signal, $m_\phi = 100$ GeV $f/c_d^{1/2} = 500$ GeV $[\lambda = 1]$ | $\nu\bar{\nu}e^- e^+$ background |
|---|---|---|
| Generation cuts | 0.084 [0.139] | 754 |
| MIM > 560 [200] GeV | 0.061 [0.139] | 311 [541] |
| $\Delta\eta_{ee} > 5.5$ [6] | 0.046 [0.106] | 3.49 [19.4] |
| $\slashed{E}_T > 80$ [80] GeV | 0.028 [0.071] | 0.472 [2.76] |
| $M_{ee} > 1100$ [2200] GeV | 0.026 [0.058] | 0.391 [0.501] |

As the derivative Higgs portal is a non-renormalizable operator, we must pay attention to follow a consistent procedure when setting limits on its coefficient. For this purpose, we adopt the method proposed in Ref. [72]: in addition to applying the selection cuts we discard events with characteristic energy $E > g_* f/c_d^{1/2}$, where $g_* \leq 4\pi$ is a coupling, because such events cannot be described within the EFT. We choose $E = M_{\phi\phi}$ and for a given value of $g_*$ we derive an exclusion region in the plane $(m_\phi, f/c_d^{1/2})$. The results are shown in the left

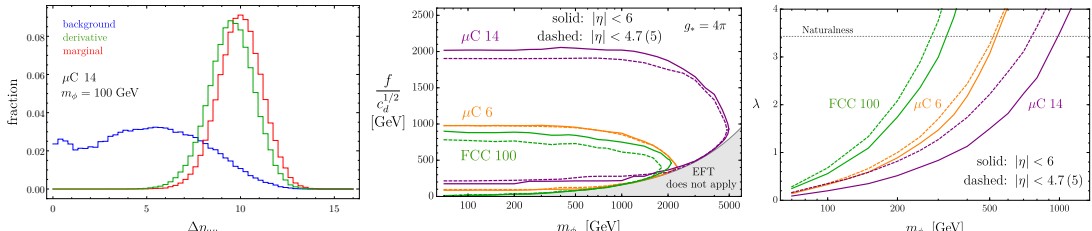

Figure 5: *Left:* normalized $\Delta\eta_{\mu\mu}$ distributions for the signals with $m_\phi = 100$ GeV and the background at a muon collider with $\sqrt{s} = 14$ TeV. Only for this plot, the generation-level cut on the muon pseudorapidity was relaxed to $|\eta_\mu| < 8$. *Middle:* projected 95% CL constraints on the derivative portal at FCC 100 and at a muon collider, assuming the forward detector coverage extends up to $|\eta| = 6$ (solid) or $|\eta| = 4.7\,[5]$ for the FCC [muon collider] (dashed). *Right:* same as the middle panel, for the marginal portal.

panel of Fig. 2, taking $g_* = 4\pi$ as an example. Projections for the marginal portal are shown in the right panel of the same figure. Unless otherwise stated, all limits are derived requiring a significance $\mathcal{S} \equiv S/\sqrt{S+B} = \sqrt{2}\,\mathrm{erf}^{-1}(1-2p)$ (Gaussian approximation, one-sided test), where for 95% CL ($p = 0.05$) the right-hand side equals 1.64.

The analysis at a future muon collider is similar, except that at the very high center of mass energies under discussion for such a machine [73] (we take $\sqrt{s} = 6$ and 14 TeV as benchmarks, see Table 1), the muons produced by the signal have extremely large pseudorapidity. The problem is most severe for $\sqrt{s} = 14$ TeV, illustrated in the left panel of Fig. 5. If the detector coverage is limited to $|\eta_\mu| < 5$ (corresponding to $\Delta\eta_{\mu\mu} < 10$) an $O(1)$ fraction of the signal is lost, causing a significant degradation of the sensitivity. Extending the detector capability in the forward region to $|\eta_\mu| < 6$ removes this problem, capturing almost all the signal with a very mild increase in background. The resulting gain in sensitivity is shown in the middle and right panels of Fig. 5. The muon collider projections in Fig. 2 assume the extended coverage $|\eta_\mu| < 6$.

## 3.2 Hadron colliders

To obtain the cross section at hadron colliders we convolute Eq. (8) with the parton luminosities in the proton,

$$\sigma(pp \to \phi\phi jj) = \sum_{q_1, q_2} \int_{4m_\phi^2/S}^1 dT \, L_{q_1 q_2}(T, Q)\, \sigma(q_1 q_2 \to \phi\phi q_1' q_2')[TS], \qquad (16)$$

with

$$L_{q_1 q_2} = \int_T^1 \frac{dx}{x}[q_1(x)q_2(T/x)]_Q\,, \qquad (17)$$

where $\sqrt{S}$ is the collider energy, $Q$ the factorization scale, and the sum in Eq. (16) extends over all relevant combinations of quarks and antiquarks. For $m_\phi \gg m_V/2$ this expression provides a sensible estimate of the cross section: we have checked that already for $m_\phi = 100$ GeV the agreement with the exact computation is within 30%, improving to $< 10\%$ for $m_\phi > 200$ GeV.

Our HL-LHC analysis is an extension of the recent searches for VBF-produced, invisibly-decaying Higgs at CMS [52] and ATLAS [76]. We generate events at parton level in MadGraph5_aMC@NLO, shower them with Pythia8 [74] and pass them through fast detector simulation with the help of Delphes3 [75], using the CMS card. The main backgrounds

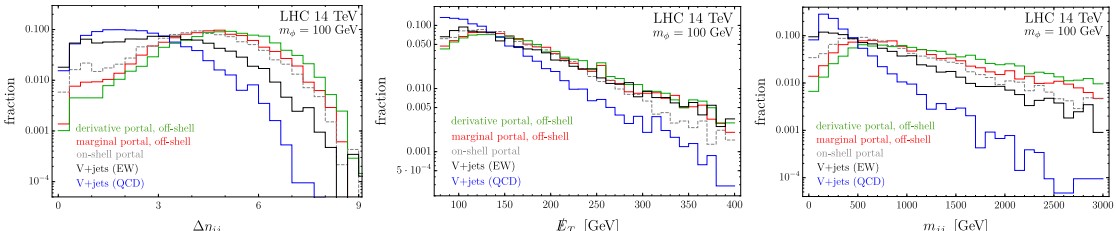

Figure 6: Normalized distributions at the 14 TeV LHC for the signals with $m_\phi = 100$ GeV and the main backgrounds, after the baseline cuts. For reference we also show normalized distributions for the on-shell signal.

Table 5: Cross sections in fb. The baseline cuts are given in Eq. (18). For 3 ab$^{-1}$ we have $\mathcal{S} = 0.15\,[0.38]$ for the derivative [marginal] portal. A dash indicates that our MC statistics is insufficient to estimate the cross section. No EFT consistency condition is applied.

| HL-LHC | signal, $m_\phi = 100$ GeV $f/c_d^{1/2} = 500$ GeV $[\lambda = 1]$ | $Z_{\nu\nu}$ QCD | $W_{\ell\nu}$ QCD | $Z_{\nu\nu}$ EW | $W_{\ell\nu}$ EW | $t\bar{t}$ |
|---|---|---|---|---|---|---|
| Baseline cuts | $0.164\,[0.618]$ | $1.4 \cdot 10^4$ | $1.7 \cdot 10^4$ | $330$ | $330$ | $1.7 \cdot 10^3$ |
| $\Delta\eta_{jj} > 4.8\,[4.2]$ | $0.090\,[0.366]$ | $440\,[960]$ | $890\,[1700]$ | $46\,[78]$ | $50\,[80]$ | $12\,[29]$ |
| $\not{E}_T > 180$ GeV | $0.028\,[0.110]$ | $58\,[140]$ | $38\,[100]$ | $14\,[26]$ | $7.8\,[15]$ | $1.5\,[4.4]$ |
| $m_{jj} > 1.97\,[1.30]$ TeV | $0.015\,[0.075]$ | $12\,[56]$ | $5.0\,[32]$ | $7.4\,[18]$ | $3.9\,[11]$ | $-\,[1.5]$ |

are $Z_{\nu\nu}$+jets and $W_{\ell\nu}$+jets. Each of them is separated into a QCD part, where the jets arise from strong interactions, and an electroweak (EW) part, where the jets are emitted purely via weak couplings. The interference between the two components is negligible [76]. Despite the smaller total cross section, the EW contributions are important because their kinematics closely resembles that of the signal, so that they constitute an $O(1)$ fraction of the total background after all cuts are applied. For the QCD processes we generate $V + 2$ jets at NLO, whereas for the EW contribution we use a LO matched sample of $V + 2, 3$ jets. We fix the overall normalization of each sample by comparing our 13 TeV predictions with the expected event yields of the CMS shape analysis, reported in Table 3 of Ref. [52]. After this rescaling the shape of all samples agrees within 10% with the CMS expectation, as shown by the left panel of Fig. 10 in Appendix A. We then apply the same rescaling factors to our 14 TeV samples. We also include the $t\bar{t}$+jets background, which is generated at LO matched with up to 2 additional partons, and normalized to the NNLO+NNLL cross section of 974 pb [77].

The baseline cuts we adopt in our 14 TeV analysis are

$$\not{E}_T > 80 \text{ GeV}, \qquad N_j \geq 2, \qquad p_T^{j_1, j_2} > 50 \text{ GeV}, \qquad |\eta_{j_1, j_2}| < 4.7, \qquad \eta_{j_1} \cdot \eta_{j_2} < 0,$$

$$N_\ell = 0, \qquad N_j^{\text{central}} = 0, \qquad \Delta\phi(j_1, j_2) < 2.2, \qquad \Delta\phi(\vec{\not{p}}_T, j) > 0.5. \tag{18}$$

The lepton veto is implemented as in the CMS analysis [52]. The central jet veto forbids events where an additional jet satisfies $p_T^j > 30$ GeV and $\min \eta_{j_1, j_2} < \eta_j < \max \eta_{j_1, j_2}$, while the $\Delta\phi(\vec{\not{p}}_T, j)$ requirement is applied to any jet with $p_T^j > 30$ GeV. Normalized distributions for the signal and background after the initial cuts in Eq. (18) are shown for $m_\phi = 100$ GeV in Fig. 6 (for comparison, in the same figure we also show normalized distributions for the on-shell Higgs signal, which are independent of the type of portal). An important discriminating variable is $\Delta\eta_{jj}$, which is most effective for the derivative portal (Fig. 6-left). This is the opposite of what happens at lepton colliders, where the $\Delta\eta_{\ell\ell}$ distribution is harder for

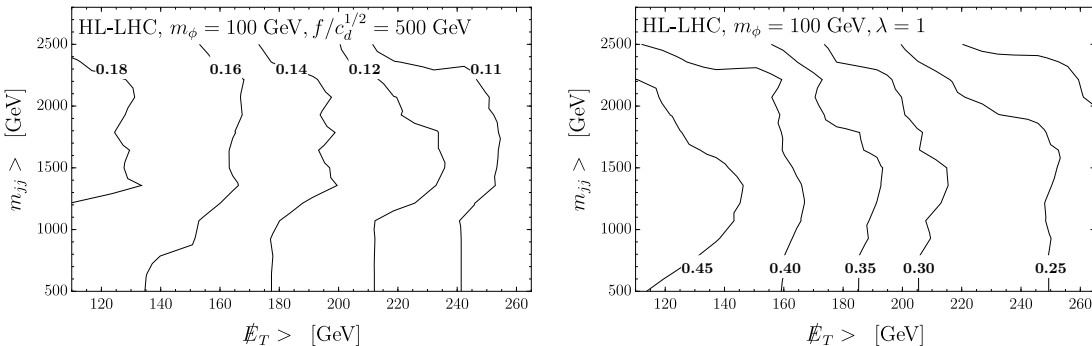

Figure 7: Isocontours of the significance $\mathcal{S}$ in the plane of the $(\not{E}_T, m_{jj})$ cuts, at the HL-LHC. Baseline and $\Delta\eta_{jj}$ cuts have already been applied.

the marginal portal, see e.g. the left panel of Fig. 5. We fix $\Delta\eta_{jj} > 4.8$ [4.2] for the derivative [marginal] portal. Additionally, we consider a more stringent cut on $\not{E}_T$ and a cut on $m_{jj}$. The significance obtained by varying these two requirements is shown in Fig. 7. It is apparent that a relatively mild cut on $\not{E}_T$ would be preferred for both portals. However, as events must be selected using a missing energy trigger, a rather stringent requirement on $\not{E}_T$ needs to be imposed. Following the ATLAS analysis [76] we set $\not{E}_T > 180$ GeV. Note that the shape of the missing energy distribution is very similar for the signals and the EW backgrounds (middle panel of Fig. 6). Finally, we optimize the $m_{jj}$ cut, separately for the derivative and marginal signals. The complete cut flow for the benchmark $m_\phi = 100$ GeV is reported in Table 5. The $t\bar{t}$+jets background is small, and we neglect it in our HL-LHC projections. Note that the EW $V$+jets are very important, making up 40% of the total background to the derivative signal. For different $m_\phi$ hypotheses all cuts are kept identical, except for $m_{jj}$, which is optimized in each case.

It is interesting to check the effect on the significance of varying the missing energy cut. We take as examples $\not{E}_T > 150$ GeV, which at ATLAS corresponds to a trigger efficiency of $\sim 90\%$ on signal events [78] (to be compared with 98% for the reference choice $\not{E}_T > 180$ GeV [76]) and $\not{E}_T > 250$ GeV, as required in the CMS analysis [52]. For the derivative portal with $m_\phi = 100$ GeV the bound on $f/c_d^{1/2}$ varies from 290 to 260 GeV, whereas for the marginal portal with naturalness-inspired coupling strength the reach on $m_\phi$ varies from 135 to 117 GeV.

Our HE-LHC analysis proceeds along similar lines. We apply to the $V$+jets backgrounds the same rescaling factors derived from the comparison with CMS at 13 TeV, whereas $t\bar{t}$+jets is normalized to the theoretical prediction of 3.73 nb [79]. For Delphes3 we use the HL-LHC card. The baseline cuts are as in Eq. (18), except that we impose $|\eta_{j_1,j_2}| < 4.9$. As additional requirements we apply $\Delta\eta_{jj} > 5.2$ [4.2] for the derivative [marginal] signal, fix $\not{E}_T > 200$ GeV, and as before we optimize the $m_{jj}$ cut for each $m_\phi$ and portal hypothesis.

For the FCC analysis we again apply the 13 TeV rescaling factors to the $V$+jets backgrounds, whereas $t\bar{t}$+jets is normalized to 34.7 nb [80]. We use the FCC Delphes card. The baseline cuts are the same as in Eq. (18), except for the requirement on the jet pseudorapidity, which is increased to $|\eta_{j_1,j_2}| < 6$. As emphasized in the FCC-hh Conceptual Design Report [81], extending detector coverage up to $|\eta_j| = 6$ would be important for the measurement of VBF processes. This is quantified for our signal in the middle and right panels of Fig. 5, where we show the gain in sensitivity obtained extending the forward jet measurement from the LHC range $|\eta_j| < 4.7$ to the expected FCC design $|\eta_j| < 6$ (note that the FCC 100 bounds shown in Fig. 2 assume coverage up to $|\eta_j| = 6$). Additionally, we set $\Delta\eta_{jj} > 5.5$ [4.2] for the derivative [marginal] portal, whereas the missing energy cut is fixed to $\not{E}_T > 200$ GeV [55]. The complete cut flow for $m_\phi = 100$ GeV is given in Table 6. For the marginal portal the

$t\bar{t}$+jets background is not negligible, hence we consistently include it in our FCC projections.

Table 6: Cross sections in pb. The baseline cuts are in Eq. (18), except that $|\eta_{j_1, j_2}| < 6$. For 30 ab$^{-1}$ we have $\mathcal{S} = 1.0\,[5.2]$ for the derivative [marginal] portal. A dash indicates that our MC statistics is insufficient to estimate the cross section. No EFT consistency condition is applied.

| FCC 100 | signal, $m_\phi = 100$ GeV $f/c_d^{1/2} = 1$ TeV $[\lambda = 1]$ | $Z_{\nu\nu}$ QCD | $W_{\ell\nu}$ QCD | $Z_{\nu\nu}$ EW | $W_{\ell\nu}$ EW | $t\bar{t}$ |
|---|---|---|---|---|---|---|
| Baseline cuts | $7.6\,[114]\cdot 10^{-4}$ | 170 | 220 | 3.5 | 3.7 | 49 |
| $\Delta\eta_{jj} > 5.5\,[4.2]$ | $5.9\,[90]\cdot 10^{-4}$ | $12\,[38]$ | $24\,[64]$ | $0.79\,[1.5]$ | $0.97\,[1.9]$ | $2.0\,[6.6]$ |
| $\not{E}_T > 200$ GeV | $2.1\,[29]\cdot 10^{-4}$ | $1.7\,[6.0]$ | $1.6\,[5.4]$ | $0.24\,[0.51]$ | $0.17\,[0.37]$ | $0.11\,[0.50]$ |
| $m_{jj} > 6.62\,[2.30]$ TeV | $0.87\,[17]\cdot 10^{-4}$ | $0.062\,[1.3]$ | $0.060\,[1.3]$ | $0.053\,[0.29]$ | $0.032\,[0.22]$ | $-\,[0.12]$ |

A comment is in order about the role of systematic uncertainties, which have thus far been neglected in our analysis. At lepton colliders the $S/B$ ratio is at least several per-cent (see e.g. Table 4), therefore the effect of systematics is expected to be minor. By contrast, at hadron colliders the $S/B$ is below the per-mille, as can be read in Tables 5 and 6, hence systematics will have an important impact on our hadron collider limits. We quantify this by repeating the HL-LHC analysis including a 1% systematic uncertainty on the total background (see e.g. Ref. [82] for the theoretical advancement in the precise predictions of the dominant $V$+jets backgrounds). The selection remains the same as in Table 5, except for the cut on $m_{jj}$, which is driven to larger values by the need to suppress the background to the strongest extent possible, as can be seen in the middle panel of Fig. 10. The resulting limits are shown by the gray bands in Fig. 2. A more comprehensive discussion of systematics lies beyond the scope of this first study.

## 4 Discussion and outlook

We begin our discussion with a few comments about Fig. 2. An important result is that hadron colliders have stronger sensitivity to the derivative Higgs portal than to the marginal Higgs portal, because the former produces harder kinematic distributions that allow a more effective background suppression (see also Tables 5 and 6). To make this statement more quantitative, we show in Fig. 8 the bounds on the (absolute value of the) effective coupling strength at the kinematic threshold $M_{\phi\phi} = 2m_\phi$, which is $c_d 4m_\phi^2/f^2$ and $\lambda$, respectively. For hadron colliders the threshold is a reasonable point of comparison, due to the PDF suppression at higher $M_{\phi\phi}$. We find a better reach on the derivative portal, thanks to the tail at higher invariant masses. For $m_\phi = 100$ GeV, the ratio of the effective coupling bounds is $\sim 1/4.0$ at the HL-LHC, $\sim 1/5.5$ at the HE-LHC and $\sim 1/11$ at the FCC. A consequence of this increased sensitivity is that, for example, FCC 100 gives projected constraints comparable to a 6 TeV muon collider for the derivative coupling, but very similar to CLIC 3 for the marginal coupling.

Our key motivation to consider the derivative portal is pNGB DM. To assess the future prospects we follow the relic density contours in the left panel of Fig. 3, where we read that $m_\chi > m_h/2$ corresponds to $f \gtrsim 500$ GeV (for complex DM). This region is out of the reach of HL-LHC and CLIC 1.5, can be probed to a very limited extent at the HE-LHC and CLIC 3, and would be truly explored only at FCC 100 or a muon collider. In particular, a $\mu$C 14 would test DM masses up to about 600 GeV.

As a representative example of the marginal portal, we consider scalar top partners with

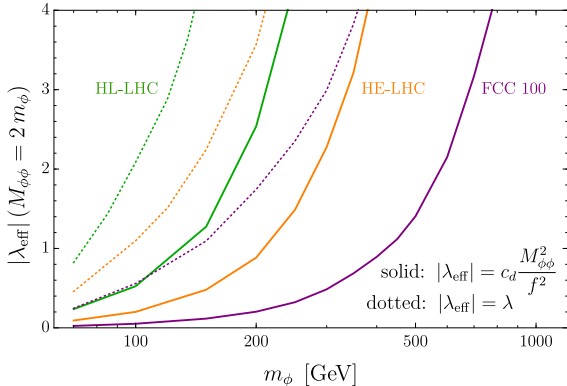

Figure 8: Hadron collider sensitivity on the effective coupling evaluated at the threshold $M_{\phi\phi} = 2m_\phi$, for the derivative and marginal portals, as obtained from our analysis. The figure quantifies the sensitivity gain that follows from the relative scaling $\propto \hat{s}^2$ in Eq. (12).

coupling fixed by naturalness. In this case even FCC 100 will have a limited sensitivity, probing top partner masses up to approximately 300 GeV. A muon collider would have a clearly superior reach, extending all the way up to $m_{\tilde{t}^c} \approx 1$ TeV for center of mass energy of 14 TeV. This would constitute an impressive, model-independent test of the possibility that the little hierarchy is stabilized by scalar top partners, regardless of their decay pattern.

We conclude with some comments about other potential probes of the pNGB DM scenario. Firstly, in this setup the couplings of the Higgs to visible particles are generically expected to show deviations from the SM predictions. The size and pattern of the corrections is, however, dependent on the model (and in particular, on whether the Higgs also arises as a pNGB or not), and their discussion is outside the scope of this paper. As far as probes of the pNGB DM itself are concerned, we emphasize that the strategy followed in this first study is simply the most direct one, exploiting the tree-level production of a DM pair through an off-shell Higgs. At hadron colliders an alternative avenue could be provided by the mono-Higgs signature [83], which allows to exploit gluon fusion production, thus partly compensating for the heavier mass threshold. In addition, probes of virtual effects may provide competitive sensitivity, in analogy to the one-loop tests of the marginal Higgs portal proposed in Refs. [84–86]. We believe the study of such indirect sensitivity to pNGB DM deserves further attention [87].

# Acknowledgments

We have benefited from conversations with F. Bishara, R. Frederix, J. Herrero-García and P. Panci. We thank S. Hong for correspondence about Ref. [44], and A. Tesi for clarifications about Ref. [69]. We also thank M. J. Ramsey-Musolf for comments about v2. This research has been partially supported by the DFG Cluster of Excellence 153 "Origin and Structure of the Universe" and by the Collaborative Research Center SFB1258. MR is supported by the Studienstiftung des deutschen Volkes, and acknowledges the hospitality of the Cornell theory group in the final stages of the project. All authors warmly thank the MIAPP, where part of this work was done.

# Appendices

# A  Details of the analysis and comparison with previous works

At lepton colliders we cut on four kinematic variables, namely MIM, $\Delta_{\ell\ell}$, $\not{E}_T$ and $M_{\ell\ell}$:

- The cuts on the MIM are summarized in the left-most panel of Fig. 9 for the derivative portal. For the marginal portal we always take MIM $> 2m_\phi$, and in a few cases we also impose an upper bound: at ILC 1, MIM $< \{175, 280, 300, 380, 450\}$ GeV for $m_\phi = \{70, 85, 100, 120, 150\}$ GeV; at CLIC 1.5 and 3, MIM $< \{250, 300\}$ GeV for $m_\phi = \{70, 85\}$ GeV.

- The cuts on $\Delta\eta_{\ell\ell}$ are shown in the middle-left panel of Fig. 9. For the marginal portal at ILC 1 this cut is replaced by $H_T(ee) < 260$ GeV, as in Ref. [44].

- The cuts on $\not{E}_T = $ MET are shown in the middle-right panel of Fig. 9 (the points that are not immediately visible in this plot all have $\not{E}_T > 80$ GeV, e.g. the marginal portal at CLIC 3).

- The cuts on $M_{\ell\ell}$ are shown in the right-most panel of Fig. 9.

For hadron colliders, we show in the middle panel of Fig. 10 the cuts on $m_{jj}$.

## A.1  Comparison to Ref. [48]

We deem it useful to provide a thorough comparison of our results for the marginal portal at hadron colliders to those of Ref. [48]. For this purpose we have tried to reproduce their VBF analysis as closely as possible, given the available information. The results are shown in the right panel of Fig. 10, together with the bounds quoted in Ref. [48] and those obtained in our main analysis.[11] We are able to reproduce well the results of Ref. [48], agreement being excellent for the FCC and good for the HL-LHC. This exercise helps us to pinpoint the differences between our analysis and that of Ref. [48]:

1) In Ref. [48] the EW components of $V$+jet were neglected, while the QCD components were generated at LO and normalized to the MadGraph5 cross sections. In this work we have included EW production and generated the QCD part at NLO; normalizations were fixed by comparison with the 13 TeV CMS results [52].

2) In Ref. [48] both the $\not{E}_T$ and $m_{jj}$ cuts were optimized, without imposing a trigger-motivated lower limit on the former. As a result the optimal values were $\not{E}_T \gtrsim 120$ GeV at the HL-LHC and $\not{E}_T \gtrsim 100$-$160$ GeV at the FCC. In this work we have fixed $\not{E}_T > 180$ and 200 GeV, respectively, and optimized only the $m_{jj}$ cut.

3) In our FCC analysis we have assumed forward jet tagging up to $|\eta| = 6$, to be compared with 4.7 used in Ref. [48].

Owing to the points 1) and 2) our HL-LHC sensitivity is weaker, while at the FCC the point 3) more than compensates for the first two, allowing us to derive slightly more stringent limits.

---

[11]Only for this plot, following Ref. [48] we adopt the coupling normalization $c_\phi = \lambda/2$ and use a two-sided test to determine the exclusion bounds, i.e. we require $\mathcal{S} = \sqrt{2}\,\mathrm{erf}^{-1}(1-p) \approx 1.96$ for $p = 0.05$ (95% CL).

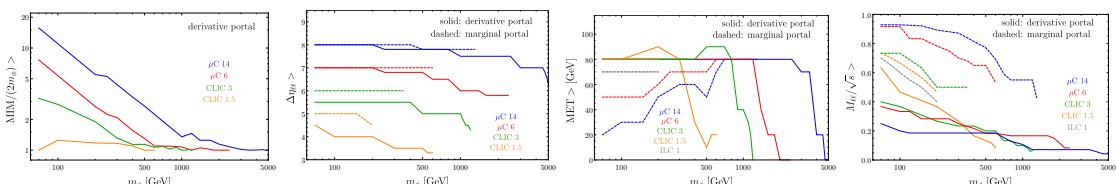

Figure 9: Summary of the cuts we apply at lepton colliders. The cuts not shown here are discussed in the text.

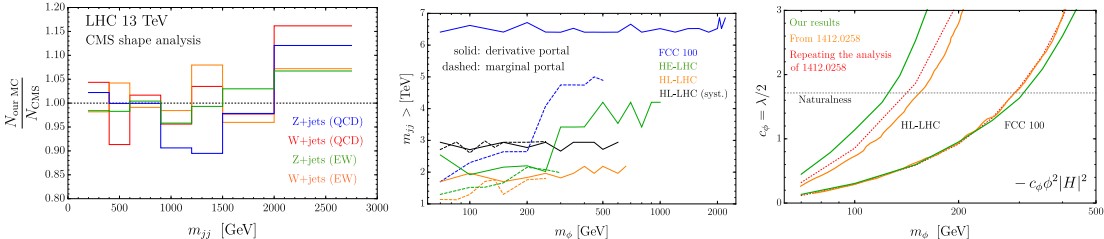

Figure 10: *Left:* comparison of our MC predictions to the expected yields in the CMS shape analysis (reported in Table 3 of Ref. [52]). Each of our MC samples has been normalized to the total expected yield quoted by CMS. The required rescaling factors are $1.18, 1.08$ for $Z_{\nu\nu}(\text{QCD}), W_{\ell\nu}(\text{QCD})$, and $1.74, 1.63$ for $Z_{\nu\nu}(\text{EW}), W_{\ell\nu}(\text{EW})$. We do not show the $m_{jj} \in [2.75, 3.5]$ TeV and $m_{jj} > 3.5$ TeV bins because there the statistical uncertainty of our MC is large. *Middle:* summary of the optimized $m_{jj}$ cut we adopt at hadron colliders. For the HL-LHC with 1% systematic uncertainty we take $m_{jj} > 3$ TeV as the strongest possible cut, due to the limited MC statistics. *Right:* comparison of our results for the marginal portal at hadron colliders with those of Ref. [48]. See text for further details.

# B  Cross sections for pNGB Dark Matter

To calculate the cross section for DM scattering on nuclei, we first match Eq. (5) to the following effective DM-SM interactions,

$$\sum_{\psi=u,d,c,s,t,b} \Big( m_\psi a_\psi |\chi|^2 \overline{\psi}\psi + i(\chi^* \overset{\leftrightarrow}{\partial_\mu} \chi) \overline{\psi}\gamma^\mu (\lambda_\psi^V$$

$$+ \lambda_\psi^A \gamma_5)\psi\Big) + \frac{ie}{m_*^2} c_B (\chi^* \overset{\leftrightarrow}{\partial_\mu}\chi)\partial_\nu F^{\mu\nu} + \frac{d_G\, y^2}{16\pi^2}\frac{g_s^2}{m_*^2}|\chi|^2 G_{\mu\nu}^a G^{a\mu\nu}, \qquad (19)$$

with coefficients

$$a_\psi = \frac{\lambda}{m_h^2} + \frac{c_\psi^\chi}{f^2} \qquad \lambda_\psi^V = \frac{b_{\psi_R} + b_{q_L}}{2f^2}, \qquad \lambda_\psi^A = \frac{b_{\psi_R} - b_{q_L}}{2f^2}. \qquad (20)$$

From these we derive the spin-independent DM-nucleon cross section up to one loop,

$$\sigma_{\text{SI}}^{\chi N, \chi^* N} = \frac{1}{\pi}\left(\frac{m_\chi m_N}{m_\chi + m_N}\right)^2 \frac{1}{A^2}\Bigg\{ Z\Bigg[ \frac{m_p}{2m_\chi}\Big( \sum_{\psi=u,d,s} f_{T_\psi}^p a_\psi + \frac{2}{27}f_{T_g}^p\Big( \sum_{\psi=c,b,t} a_\psi - 3\frac{d_G\, y^2}{m_*^2}\Big)\Big)$$

$$\pm \sum_{\psi=u,d} f_{V_\psi}^p (\lambda_\psi^V + \delta\lambda_\psi^V)\Bigg] + \{p \to n, Z \to A - Z\}\Bigg\}^2, \qquad (21)$$

where $\delta\lambda^V_\psi = c_B e^2 Q_\psi / m_*^2$ and $m_N = (m_p + m_n)/2$. For the scalar operators, we have separated the contribution of the light quarks from that of the heavy quarks. The associated form factors take the values (see App. D of Ref. [15] for the relevant references) $f^p_{T_u} = 0.021, f^p_{T_d} = 0.041,$ $f^n_{T_u} = 0.019, f^n_{T_d} = 0.045,$ and $f^{p,n}_{T_s} = 0.043,$ leading to $f^{p,n}_{T_g} = 1 - \sum_{\psi=u,d,s} f^{p,n}_{T_\psi} \simeq 0.89.$ It is important to note that for the vector current operators proportional to $\lambda^V_\psi$, the form factors simply count the valence content of the nucleons, namely $f^p_{V_u} = f^n_{V_d} = 2$ and $f^p_{V_d} = f^n_{V_u} = 1$. The additional term proportional to $\delta\lambda^V_\psi$ originates from $t$-channel photon exchange.

As for DM annihilation, the last two operators in the first line of Eq. (5) contribute to the $p$-wave, while all other operators contribute to the $s$-wave. For $m_\chi \gtrsim m_W$ the $\chi\chi^* \to WW, ZZ,$ $hh$ channels dominate and the $b_\Psi$ and $c_B$ operators are subleading to the unsuppressed Higgs portal. Conversely, for $m_\chi \lesssim m_W$ the $\chi\chi^* \to \psi\overline{\psi}$ channel (where $\psi$ is a SM fermion) dominates and the $b_\Psi$ and $c_B$ operators are important at freeze-out, when their velocity suppression can be less severe than the $m_\psi$-suppression that characterizes the other operators. As an example we consider $\chi\chi^* \to b\bar{b}$, for which we find a cross section

$$\sigma(\chi\chi^* \to b\bar{b})v_{\rm rel} =$$
$$\frac{N_c}{4\pi s}\sqrt{1 - \frac{4m_b^2}{s}}\left\{A^2(s)m_b^2(s - 4m_b^2) + \frac{2}{3}(s - 4m_\chi^2)\left[B^2(s)(s + 2m_b^2) + C^2(s)(s - 4m_b^2)\right]\right\},$$
(22)

with

$$A = \left(\frac{c_d s}{f^2} - \lambda\right)\frac{c_{hbb}}{s - m_h^2} + \frac{c^\chi_{d3}}{f^2},$$
(23)

$$B = \frac{c_B}{m_*^2}\left(e^2 Q_b - \frac{g_Z^2 s_w^2 v_b s}{s - m_Z^2}\right) + \frac{b^3_{q_L} + b^3_{d_R}}{2f^2},$$
(24)

$$C = -\frac{c_B}{m_*^2}\frac{g_Z^2 s_w^2 a_b s}{s - m_Z^2} + \frac{b^3_{q_L} - b^3_{d_R}}{2f^2},$$
(25)

where $c_{hbb} = g_{hbb}/g^{\rm SM}_{hbb}$ and the coupling of the SM fermion $\psi$ to the $Z$ is defined as $ig_Z\gamma^\mu(v_\psi - a_\psi\gamma_5)$.
Taking the thermal average and considering for simplicity the limit $m_\chi \gg m_b$ we arrive at

$$\langle\sigma v_{\rm rel}\rangle(T) \simeq \frac{N_c}{4\pi}\left\{m_b^2 A^2(4m_\chi^2) + 4m_\chi T\left[B^2(4m_\chi^2) + C^2(4m_\chi^2)\right]\right\}.$$
(26)

At the freeze-out temperature $\sim m_\chi/25$, for $A^2 \sim B^2 + C^2$ we have parametrically

$$\frac{\langle\sigma v_{\rm rel}\rangle_{p\text{-wave}}}{\langle\sigma v_{\rm rel}\rangle_{s\text{-wave}}} \sim \frac{4}{25}\frac{m_\chi^2}{m_b^2}.$$
(27)

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
