# Peer review of "A Global View of the Off-Shell Higgs Portal"

_SciPost Physics, doi:SciPost Phys. 8, 027 (2020)_

## Round 1 · Referee Report · Anonymous (Referee 1) · 2019-11-18

Report

In their work, the authors derive the constraints that future measurements of dark matter pair production in the vector-boson-fusion Higgs production channel may allow to set on the derivative and the marginal Higgs portal. While the marginal Higgs portal has already been studied before the analysis of the derivative Higgs portal is new. The paper is clearly written, the analysis described in detail and the results (while a bit depressing) are interesting. In view of this, I think that the work should be published as is. Below I list a couple of optional changes that in my opinion would improve the quality of the manuscript even further.

Requested changes

1) Some of the plots are too small for my taste (for instance Figure 9). The colours in Figure 6 that indicate the "V+jets (EW)" background and the "marginal portal" are also hard to disentangle. I would suggest that the authors try to improve this.

2) In footnote 9 the authors write "... the second ..." which I guess should read "... the second term ...". The same statement applies to "... first ..." and "... in the last ...".

3) I have spotted the following typos/grammatical mistakes in the manuscript. a) "The hypothesis that (the) Dark Matter", b) "... as THE low-energy theory ...", c) "... through AN off-shell Higgs ...", d) "We discuss our results and GIVE AN outlook ...", and d) "... as AN example ...". In addition I would suggest to introduce the abbreviation "SILH" and write "right-hand side" instead of "RHS".

---

## Round 1 · Referee Report · Anonymous (Referee 2) · 2019-12-5

Strengths

1- the paper studies a novel and very interesting model at colliders; 2- it covers all different collider scenarios and cosmological constrains, making it very comprehensive

Weaknesses

1- the structure of the paper is weird, with its executive summary in front, the long and not too interesting model description, and covering FCC before HE-LHC; 2- I fear the numbers are hardly reproducible, which is a problem given that the simulation is sometimes rather naive and definitely not at the level where experiments can just run with it; 3- I am a little unsure about the bottom line, probably Fig.2, but the left and right panels are hard to combine to a coherent picture.

Report

The idea behind the paper is very cute, and it is extremely interesting. Indeed, people have looked at the simple-minded invisible Higgs decays through the renormalizable (most people I know do not call it marginal) Higgs portal too much. The issue is that there are lots and lots of VBF Higgs to invisible papers, not many mentioned here, and people will want to compare renormalizable to EFT results. Some of this is possible from the paper, but not enough. Sometimes it is also just the layout which makes it hard, like the two y axes in Fig.2 which cannot be matched to each other (in Fig.8 this is possible). So my bottom line is that the paper is interesting, but it is not presenting the results in a way that is especially useful. Essentially, the paper is asking other groups to re-do the analysis for more details, and that is not all too useful. Of course I am just the referee and not an author, but maybe the authors tried to make the paper shorter than it deserves to be with all the different studies, and this way too much information got lost? Happy holidays!

Requested changes

1- see above weaknesses; 2- most importantly, please provide the renormalizable portal results in a way that ATLAS/CMS can compare to their studies. They typically use BR, not lambda; 3- please be more careful with the references, lots of stuff missing, like DM limits on the Higgs portal, ZH production at the LHC, original Higgs-portal DM, unitarization of VBF. And in some cases there might be also some of the related content missing; 4- along a similar line, what does the left panel of Fig.3 tell us, really? How should we understand this model, with a non-renormalizable operator, in terms of possible UV completions and cosmological implications; 5- from Ref.[37] I am missing the maybe central analysis tool, the CJV. Many ATLAS/CMS do not use it, either, or they replace it by something, but this just means that they are poorly done and will be changed; 6- make sure the hadron collider results can, for the renormalizable portal, be compared to other studies in the literature, for instance the different HL-LHC, HE-LHC, and FCC reports. This is crucial to judge how realistic the new results are, and it is easy to do; 7- similarly, how do the ILC and FCC results compare to the literature when translated into a renormalizable Higgs portal BR; 8- really, the argument for ignoring systematics is about the least elegant way of saying `we could not be bothered'. I think it is fine to not be bothered, but please be honest about it or find a better pseudo-argument. Systematics challenges are at least as well known as the luminosities of those future colliders...

  • validity: good
  • significance: high
  • originality: top
  • clarity: ok
  • formatting: good
  • grammar: excellent

Author:  Ennio Salvioni  on 2019-12-11  [id 675]

(in reply to Report 2 on 2019-12-05)

As the first and most important point, it is senseless to ask for our limits to be presented using branching ratios. The focus of our paper is invisible \phi scalar pair production through off-shell Higgs, m_\phi > m_h/2. As a consequence, it is simply not meaningful to formulate our bounds on \lambda as limits on BR(h -> inv), as the referee repeatedly asks in the Requested changes #2, #6 and #7. There is no branching ratio as the Higgs cannot decay on-shell into this final state. The failure to understand this central point, shows that the referee has probably not spent enough time to understand the basic idea of the paper (which should be clear from its title, where `off-shell' appears!).

A second key point concerns the supposed Weakness #2: we find the referee's allegation that our numbers would be `hardly reproducible,' without any arguments supporting such statement, frankly insulting. To the contrary, we have made a special effort to provide details on our signal and background simulations and to show all cuts adopted in the analyses, which are given in Appendix A. While our analyses could potentially be improved by implementing advanced techniques dedicated to each collider option (which is beyond the scope of this overview), we believe their results are solid and, most importantly, easily reproducible using the information provided in the paper.

Concerning Weakness #3, the results summarized in the left and right panels of Figure 2 do not necessarily need to be combined into a coherent picture, as the two types of portals can arise from different ultraviolet physics. Nevertheless, we did provide a comparison of the sensitivity on the two types of interaction at hadron colliders, as shown in Figure 8 and discussed in the first paragraph of Section 4. As a general remark, we would appreciate it if the referee could suggest concrete steps or point to actual information missing from the paper, instead of issuing vague and general declarations.

Requested change #4: a brief review of the models where the derivative Higgs portal, and therefore Figure 3, are relevant is provided by Section 2, whose goal is precisely to introduce unfamiliar readers to the main ideas. Extended discussions of the composite pNGB DM scenario can be found e.g. in the original work of Ref. [2] (Frigerio, Pomarol, Riva and Urbano 1204.2808) and, for recent developments, Ref. [15] (Balkin, Ruhdorfer, Salvioni and Weiler 1809.09106).

Requested change #5: the central jet veto is actually always included in our hadron collider studies, see Eq. (18).

Turning now to minor, if useful, improvements that we will implement in v2 following the referee's suggestions:

  • In response to Weakness #1, we agree that the HE-LHC should be discussed before FCC-hh, and we will switch the order in the text.
  • It is true that \lambda is often called renormalizable Higgs portal, and we will add a mention of this when introducing Eq. (2).
  • Responding to the demand for additional references in Requested change #3, for completeness we will include citations to the original Higgs portal DM papers (Silveira and Zee 1985; McDonald 1994; Burgess, Pospelov and ter Veldhuis 2000), as well as to the recent review Arcadi, Djouadi and Raidal 1903.03616 for current experimental limits. However, we do not see the direct relevance to our work of papers on VBF unitarization or Zh production at the LHC. If the referee has specific references in mind, we ask them to provide the details so that we can consider them. Note that just below Table 4 we have described our procedure for a consistent use of the EFT, along the lines of Ref. [58].
  • In relation to Requested change #5 (see above), after Eq. (18) we will provide details on our definitions of the lepton and central jet vetoes, as well as on the \Delta\phi (\vec{\slashed{p}}_T, j) requirement.
  • About Requested change #8: our point is simply that a study of systematics goes beyond the scope of our overview, and we will modify slightly the wording of the last sentence of Section 3 to avoid misinterpretations. We will also add a comment that under this assumption, if the integrated luminosity is L' rather than our assumed L, the reach on f/\sqrt{c_d} scales as (L' / L)^{1/8} and the reach on \lambda scales as (L/L')^{1/4} (this assumes S << B, which is well verified throughout). In fact, we agree that the values of L we have chosen are only educated guesses.

Anonymous on 2019-12-11  [id 676]

(in reply to Ennio Salvioni on 2019-12-11 [id 675])

I am sorry if the authors are insulted by my ignorance, but I am happy to see that they added the original Higgs portal references. Given that their analysis is really a slightly modified version of a standard on-shell invisible Higgs analysis it would have been useful to benchmark this on-shell version in terms of invisible branching ratios. Advanced analysis methods will of course make a big difference for each of the colliders considered, and it would be nice to have a base line. But then, it is not my paper, it is kind of good enough as it stands, so we can as well just publish it...

---

## Round 1 · Referee Report · Anonymous (Referee 3) · 2019-12-19

Strengths

a

Report

The main message of this paper is that data obtained from hadron colliders can be used to distinguish between a renormalizable Higgs portal and a dimension 6, momentum-dependent Higgs portal operator. The authors focus on vector-boson fusion (VBF), based on the study ( https://arxiv.org/pdf/1412.0258.pdf ), which has identified VBF as the most promising channel for above threshold Higgs mediated dark matter production.
The momentum-dependent Higgs portal predicts harder kinematic distributions for the two recoil jets in VBF than the "marginal" higgs portal. Stronger cuts can be applied, yielding better sensitivity for the signal.

At the example of a specific model of pseudo-nambu goldstone dark matter, ranges for masses and coupling strengths are motivated by the relic abundance of dark matter. The authors use a monte carlo simulation for VBF signal and backgrounds(madgraph, pythia8) and for detector simulation delphes3 and compare results from LHC and FCC-hh with projections for VBF signals at ILC, CLIC and potential future muon colliders.

The paper is well-structured and clearly written. The physics is sound and the argument is relevant, since distinguishing axion-like dark matter from scalar dark matter would reveal important information about the structure of a dark matter sector. I recommend publication, but also make a number of recommendations that would improve the paper:

  1. The authors focus on tree-level processes and refer to a future publication for constraints from loop-induced couplings. It would have been interesting to at least discuss the estimated impact from indirect constraints, e.g. corrections to the shape of the Higgs-potential.

  2. The focus on VBF is justified, but based on precisely the argument the authors make other signatures could even be more relevant. E.g. the mono-Higgs signal should have contributions leading to even harder spectra (scaling like $c_d^4$) in comparison with the "marginal" portal.

  3. Systematics play an important role in the comparison of hadron with lepton colliders. The authors acknowledge this, but refer to future work instead of an analysis. It would be useful to estimate the impact and show bands for the projections of the hadron machines when they are compared (e.g. in fig.2). The limitation of lepton colliders at the energy frontier is similar to the inherent disadvantage of hadron colliders due to the less clean environment, so phrasing it as "best sensitivity that can be achieved in principle" is a bit unfair.

---

## Round 2 · Referee Report · Anonymous (Referee 5) · 2020-1-6

Report

I recommend the manuscript for publication in its present form.

---

## Round 2 · Referee Report · Anonymous (Referee 4) · 2020-1-6

Report

I recommend that the manuscript is published in its present form.

---

## Round 2 · Author Response

In addition to the changes listed below, we wish to elaborate further about four points raised by the referees:

a) Concerning the desire of Referee 2 for a comparison with the on-shell signal: We emphasize that one of the main novelties of this work is to analyze the different kinematical features of the derivative and marginal (or renormalizable) portals. This difference is characteristic of the off-shell regime and disappears for on-shell decays, in which case branching ratio limits can immediately be translated into constraints on either type of portal, as already done in Table 3 for illustration. In light of this fact, and given the very extensive (and in many cases, technically very advanced) literature that already exists about on-shell decays, we do not believe that including BR(h -> inv) limits would constitute a helpful addition to this manuscript. Nonetheless, for useful comparison we have included distributions for the on-shell signal at the LHC in Fig. 6, see change #8.

b) Regarding systematic uncertainties: While a complete analysis is beyond the scope of our work, we have addressed the comments by Referees 2 and 3 through the addition of dedicated results for the HL-LHC in Fig. 2, which serve to quantify the expected effects on hadron collider constraints, as well as through a revision of the last paragraph of Section 3. See change #4.

c) About the comment on indirect constraints made by Referee 3: We fully agree that one loop probes of the derivative Higgs portal (such as, e.g., gg -> hh, similarly to what done in Ref. [78] for the marginal portal) are an interesting avenue to pursue. However, their analysis requires dedicated work that is in part ongoing, and for which no results are yet available. Other indirect observables, such as the corrections to the couplings of the 125 GeV Higgs boson, can also be relevant in probing the models we consider. These effects are, however, strongly model-dependent, and for this reason we have refrained from discussing them in the text.

d) About the observation by Referee 1 that the plots in Fig. 9 would be too small: Given that SciPost Physics is a fully electronic journal, we believe the size of this figure is acceptable.

We thank all three referees for their reading of our manuscript and for providing insightful comments. We believe the minor revisions listed below have further improved the quality of the paper, and we are hopeful that the current v2 can be accepted for publication in SciPost Physics.

---

## Round 2 · List of Changes

We list here all the changes we have made in v2, in order of appearance in the text:

1) On page 1 we have added citations to Ref. [17] and Ref. [28], which appeared on the arXiv after the v1 of our work, with the aim to provide a comprehensive overview of the literature on pNGB DM.
2) We have corrected all the typos and made all the small language improvements suggested by Referee 1.
3) When introducing the marginal Higgs portal just above Eq. (2) we have added the alternative name "renormalizable," as well as citations to the original papers Refs. [30-32]. In addition, a few lines later we have added a citation to Ref. [33] for the current constraints on this portal.
4) Both panels of Fig. 2 have been updated and now include also HL-LHC constraints derived assuming a 1% systematic uncertainty on the total background (weaker limit boundary of the gray bands). A mention of this has been added at the end of the caption of the same figure. The discussion of systematics in the last paragraph of Sec. 3 (page 12) has been revised. In addition, the middle panel of Fig. 10 and its caption have been updated accordingly.
5) In the caption of Fig. 2 we have added further information about the ILC and FCC-ee results.
6) On page 3 we have added a sentence (starting with "Note also that if ...") about the scaling of our bounds with the integrated luminosity.
7) In Eq. (5) we have added the last term in the second line, as well as the related definitions of g_V and y in the text after the equation. These operators play a minor role in our discussion, but we have nevertheless included them for completeness. Correspondingly, footnote 6 has been extended and footnote 7 has been added. The effect of these extra operators on DM direct detection has also been included in Eqs. (19) and (21) of Appendix B. Finally, we have commented on the |\chi|^2 |H|^4 operator in the text after Eq. (5).
8) Following the wish of Referee 2 for a comparison with the on-shell topology, in Fig. 6 we have added the normalized distributions for the on-shell invisible Higgs signal, which are independent of the type of portal. Comments on this are given in the caption of the same figure and in the text after Eq. (18). We have chosen to include the comparison with the on-shell case for the LHC, which is of the most immediate experimental relevance. In addition, in Fig. 6 we have changed the color for the V+jets (EW) distributions from orange to black, following a suggestion by Referee 1.
9) Just after Eq. (18) we have provided further details on the lepton veto, the central jet veto and the \Delta\phi(\vec{\slashed{p}}_T, j) requirement.
10) As suggested by Referee 2, we have inverted the order of the HE-LHC and FCC analyses in the text (page 11).
11) On page 13 we have added a sentence mentioning the mono-Higgs signal, together with a citation to Ref. [75]. We thank Referee 3 for their interesting observation about this point.

---

## Editorial Decision

published